# Deterministic control of ferroelectric polarization by ultrafast laser pulses

Peng Chen [1✉], Charles Paillard [2], Hong Jian Zhao[1,5], Jorge Íñiguez [3,4] & Laurent Bellaiche [1✉]

Ultrafast light-matter interactions present a promising route to control ferroelectric polarization at room temperature, which is an exciting idea for designing novel ferroelectric-based devices. One emergent light-induced technique for controlling polarization consists in anharmonically driving a high-frequency phonon mode through its coupling to the polarization. A step towards such control has been recently accomplished, but the polarization has been reported to be only partially reversed and for a short lapse of time. Such transient partial reversal is not currently understood, and it is presently unclear if full control of polarization, by, e.g., fully reversing it or even making it adopt different directions (thus inducing structural phase transitions), can be achieved by activating the high-frequency phonon mode via terahertz pulse stimuli. Here, by means of realistic simulations of a prototypical ferroelectric, we reveal and explain (1) why a transient partial reversal has been observed, and (2) how to deterministically control the ferroelectric polarization thanks to these stimuli. Such results can provide guidance for realizing original ultrafast optoferroic devices.

[1] Physics Department and Institute for Nanoscience and Engineering, University of Arkansas, Fayetteville, AR 72701, USA. [2] Université Paris-Saclay, CentraleSupélec, CNRS, Laboratoire SPMS, 91190 Gif-sur-Yvette, France. [3] Materials Research and Technology Department, Luxembourg Institute of Science and Technology (LIST), Avenue des Hauts-Fourneaux 5, L-4362 Esch/Alzette, Luxembourg. [4] Department of Physics and Materials Science, University of Luxembourg, 41 Rue du Brill, L-4422 Belvaux, Luxembourg. [5]Present address: International Center for Computational Method and Software (ICCMS) and Key Laboratory of Physics and Technology for Advanced Batteries, Jilin University, 2699, Qianjin Street, Changchun 130012, China. ✉email: peng.chen.iphy@gmail.com; laurent@uark.edu

Manipulation of properties of quantum materials utilizing (GHz-THz) high-frequency light is a fascinating topic in modern solid-state physics[1–6]. It has resulted in several breakthroughs in the past decade. Examples include stimulating insulator-metal transitions[7,8], controlling magnetic domains[9–13], uncovering hidden phases[14–17], inducing superconductivity[18–21] as well as the formation of photon-dressed topological states[22–24]. Light-induced switching of ferroelectric polarization[25–34] is also among these most important achievements since it can result in novel optoferroic devices, especially ultrafast nonvolatile ferroelectric memories[35,36].

In particular, recent works proposed and demonstrated that the ferroelectric polarization can be reversed by exciting a high-frequency infrared-active phonon mode (to be denoted as auxiliary mode or auxiliary high-frequency mode in the following) that is coupled to the soft mode (which is mostly associated with this polarization), with this effect acting through an intermediate anharmonic driving force[29,37,38]. Such an indirect method stimulates coherent phonon modes and has the potential to achieve polarization switching within a few hundred femtoseconds, which will be six orders of magnitude faster than from photovoltaic effects[27,28,39]. This route was first theoretically proposed by Subedi et al.[38] and then partially realized by Mankowsky et al. in their second harmonic generation (SHG) experiment[29].

However, some experimental results contradict predictions from the theory. For instance, the model of Ref. [38] predicted a full reversal of the polarization while measurements "only" reported a transient reversal[26,29], with the reversed polarization not even reaching its equilibrium value. It is presently unclear why only a partial transient reversal was observed[29] and whether it is, in fact, possible to achieve a full reversal. A possible reason for such paucity of knowledge is that the theory of Ref. [38] may have missed important couplings[40]. Moreover, the fact that the studied model of Ref. [38] is only one-dimensional in nature also implies that some striking features may have been overlooked, especially in systems that can adopt polarization along different crystallographic directions. For instance, is it possible to make the polarization rotate rather than reverse by activating an auxiliary high-frequency mode? Can we also expect novel effects when applying such light-induced indirect method to different structural phases, each having its own direction for the ferroelectric polarization, such as rhombohedral, orthorhombic, and tetragonal states in ferroelectric perovskites?

The aim of the present study is to answer all these questions by employing an original atomistic scheme that includes the three-dimensional soft mode, the three-dimensional auxiliary high-frequency mode, and all their relevant couplings. As we will see, this atomistic approach not only reproduces a transient partial reversal analogous to the one observed in the experiments of Ref. [29], but also provides insight into the light-driven effects. It further reveals and explains how a full reversal (180° rotation) can indeed happen in some cases; and also predict a variety of light-induced phase transitions, for which polarization rotates by 60°, 71°, 90°, and 109°, as a result of a mechanism we coin here as "squeezing". The discovery of such a "squeezing" mechanism further allows us to design a strategy for ultrafast deterministic control of the polarization and even realize full reversal in a deterministic manner.

## Results

As we are looking for a universal behavior of the THz response of the resonated auxiliary high-frequency mode and its consequence on electrical polarization, the prototypical ferroelectric material KNbO₃ is chosen. It allows us to explore different ferroelectric phases with different polarization directions at different temperatures, which leads to a richer playground than one-dimensional ferroelectrics (e.g., LiNbO₃) or tetragonal ferroelectrics (e.g., PbTiO₃ that has only a single transition from cubic with no polarization to tetragonal with polarization along < 001 >). A novel effective Hamiltonian, H_{eff}, is developed for such system, and is detailed in the method section. Its degrees of freedom are vectors related to the ferroelectric soft mode (**P**), high-frequency auxiliary mode (**Q**) that is polar as well, and inhomogeneous strain (**u**) in each 5-atom unit cell, as well as, the homogenous strain ($\eta$) affecting the whole simulation supercells. Such an effective Hamiltonian does not explicitly include electronic degrees of freedom but rather takes into account the ionic displacements associated with the Q and P modes. Note that this model is similar to previous effective Hamiltonians for ferroelectric perovskites[41], except for the explicit consideration of the high-frequency polar mode **Q** and its couplings. Note also that the homogeneous electric fields applied in our simulations couple to zone-center modes that we can call "TO" (as they are the limit of the TO bands for q → 0, with q being reciprocal vectors within the first Brillouin zone), but not "LO" (that involves a discontinuity that appears for q → 0, due to long-range repulsive electrostatic forces).

**Phase diagram.** Before studying the behavior of the ferroelectric polarization under a mid-infrared pulse, let us first explore the phase diagram of KNbO₃ by running Monte Carlo (MC) simulations using our first-principle-based effective Hamiltonian on a $12 \times 12 \times 12$ supercells (we numerically found that using $10 \times 10 \times 10$ or smaller supercells are not large enough to reproduce the first-order character of the phase transitions in KNbO₃). Figure 1a, b reports the temperature behavior of the three Cartesian components of the supercells average of the **P** and **Q** modes, respectively. These figures indicate that our model can nicely reproduce the sequence of phase transitions observed in experiments, that is, from cubic to tetragonal (for which a large $P_z$ coexists with a small $Q_z$), then from tetragonal to orthorhombic (where both $P_y$ and $Q_y$ get activated and become equal to $P_z$ and $Q_z$, respectively) and finally from orthorhombic to rhombohedral (for which we have now large $P_x = P_y = P_z$ and small $Q_x = Q_y = Q_z$), when cooling down the system. Note that **P** and **Q** are of opposite signs for the tetragonal (T), orthorhombic (O), and rhombohedral (R) phases, as a direct consequence of the positive signs of some coupling parameters indicated in Table 1 and further discussed later on. Examples of predicted values for the total polarization for R, O, and T phases are 54 μC/cm² at 10 K, 48 μC/cm² at 300 K, and 42 μC/cm² at 400 K. They compare reasonably well with the experimental values of $42 \pm 4$, $32 \pm 3$, and $30 \pm 2$ μC/cm² that one can find at 230, 370, and 473 K in these three phases, respectively, in Refs. [42,43]. However, as is often the case with effective Hamiltonians[44], some predicted transition temperatures are lower than the experimental ones[43], namely 640 versus 708 K for the C-to-T transition, and 345 versus 498 K for the T-to-O transition. On the other hand, the R-to-O transition temperature is well reproduced here: 260 K for the effective Hamiltonian and 263 K in measurements. Interestingly, we also numerically found that, if the **Q** degrees of freedom are turned off, the O-phase vanishes, and only two-phase transitions occur: one from T-to-C at a lower temperature around 500 K and a second one from R-to-T around 400 K. Such fact highlights the importance of incorporating both **P** and **Q** in order to accurately model properties of KNbO₃. Note, however, that a previous model without **Q** did find an O equilibrium phase, but for some narrow temperature range (≃50 K)[45]. One can also easily imagine the development of a model having **P** as the only polar degree of freedom, but where the effect of **Q** would be to renormalize the coefficients associated with **P**.

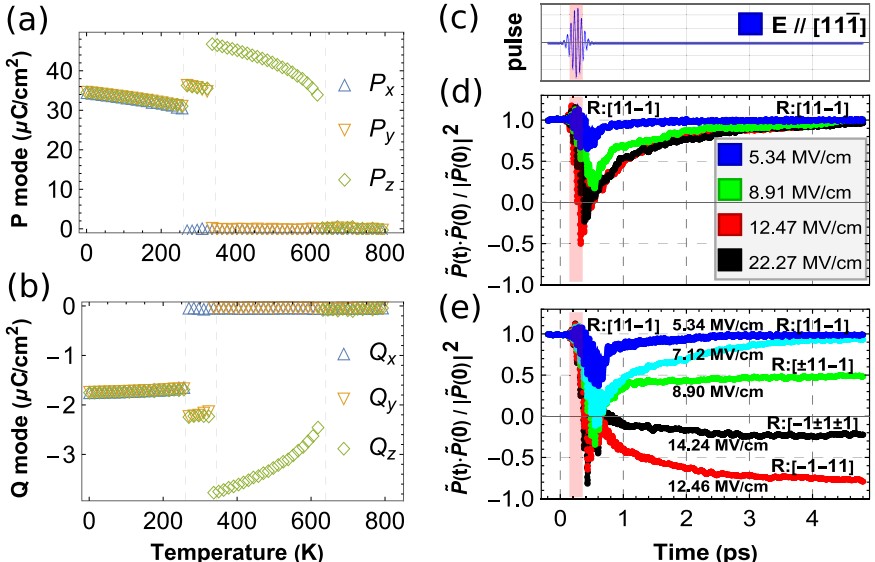

**Fig. 1 Results to be compared with experiments. P** (**a**) and **Q** (**b**) modes as a function of temperature. The ratio involving the total polarization at 240 K as a function of time, for partial-excitation (**d**) and full-excitation (**e**) starting from an R-phase. Panel **c** shares the same horizontal axes with panels **d** and **e** to indicate the laser pulse shape and time evolution. Pink regions are to indicate the full-width-half-maximum of the Gaussian enveloped laser pulse.

| Table 1 On-site couplings between ferroelectric soft mode P and high-frequency auxiliary mode Q. | |
| --- | --- |
| **Coefficients (Λ)** | **Coupling energies** |
| $2.02 \times 10^{13}$ | $\Lambda_2(P_x^2 Q_x^2 + P_y^2 Q_y^2 + P_z^2 Q_z^2)$ |
| $-2.13 \times 10^{12}$ | $\Lambda_{22}(P_x^2 Q_y^2 + P_x^2 Q_z^2 + P_y^2 Q_z^2 + P_y^2 Q_x^2 + P_z^2 Q_x^2 + P_z^2 Q_y^2)$ |
| $2.68 \times 10^{11}$ | $\Lambda_3(P_x^3 Q_x + P_y^3 Q_y + P_z^3 Q_z)$ |
| $8.86 \times 10^{11}$ | $\Lambda_1(P_x Q_x^3 + P_y Q_y^3 + P_z Q_z^3)$ |
| $9.69 \times 10^{11}$ | $\Lambda_{211}(P_x^2 P_y Q_y + P_x P_y^2 Q_x + P_x^2 P_z Q_z + P_x P_z^2 Q_x + P_y^2 P_z Q_z + P_y P_z^2 Q_y)$ |
| $-4.61 \times 10^{11}$ | $\Lambda_{112}(P_x Q_x Q_y^2 + P_y Q_x^2 Q_y + P_z Q_x^2 Q_z + P_x Q_x Q_z^2 + P_z Q_y^2 Q_z + P_y Q_y Q_z^2)$ |
| $1.70 \times 10^{13}$ | $\Lambda_{1111}(P_x P_y Q_x Q_y + P_y P_z Q_y Q_z + P_z P_x Q_z Q_x)$ |

The unit of these fourth-order coupling coefficients is $Nm^6/C^4$.

**Electrical polarization reversal**. Let us now check if the presently developed method also allows us to reproduce a situation similar to the one reported in Ref. [29], namely, an electrical polarization transient reversal in a rhombohedral state of a ferroelectric material. For that, we employ the aforementioned $H_{eff}$ within molecular dynamics (MD) simulations on an R-phase of $KNbO_3$ at 240 K, for which **P** is along [11$\bar{1}$] and **Q** antiparallel to it. Technically and in order to be close to the experimental situation of Ref. [29], we mimic the application of a "Gaussian enveloped" laser pulse of the form $\mathbf{E}e^{-2\ln 2(\frac{t}{\tau})^2}\cos(2\pi\omega t)$ (with a full-width-half-maximum (FWHM) $\tau = 200$ fs and frequency $\omega = 18$ THz) and a light polarization (**E**) that is parallel to [11$\bar{1}$]. Note also that, as detailed in Sec. I of the Supplementary Material (SM), this 18 THz frequency is chosen here because it is close to the resonance of the high-frequency auxiliary mode, while the frequency of the soft mode is found to be ≃8 THz at 240 K, according to our calculations. In order to further compare with the measurements of Ref. [29], the dot product of the total polarization at time $t$, $\widetilde{\mathbf{P}}(t) = \mathbf{P}(t) + \mathbf{Q}(t)$, and its initial value, $\widetilde{\mathbf{P}}(0)$, is calculated and then divided by the square of $|\widetilde{\mathbf{P}}(0)|$ for different magnitudes of the electric field and with the pulse starting at $t = 0$ ps. Note that, in simulations with this type of effective models, the applied electric fields are typically predicted to be 20 times larger than the experimental ones[46]. The results are shown in Fig. 1d. As only a

fraction of the material was excited in the experiment (done on a 5-mm-thick $LiNbO_3$ sample), we also assumed that only a portion of our supercell is experiencing the pulse (technically, only $10 \times 10 \times 10$ cells within the $12 \times 12 \times 12$ supercells are allowed to be coupled to the electric field) and keep the homogenous strains fixed at its initial values to mimic clamping. Only the dipoles subject to the pulse are incorporated in the computation of $\widetilde{\mathbf{P}}(0)$ and $\widetilde{\mathbf{P}}(t)$.

One can see that, for a small magnitude of the electric field associated with the pulse, e.g., 5.34 MV/cm, the polarization (blue curve in Fig. 1d) first decreases from its initial value, while still being along [11$\bar{1}$], before increasing and desiring to reach $\widetilde{\mathbf{P}}(0)$ again at larger times. The dip in polarization is enhanced when increasing the magnitude of the field to 8.91 MV/cm (green curve in Fig. 1d) until such dip becomes negative for 12.47 MV/cm (red curve in Fig. 1d). In other words, for strong enough pulses, the total polarization $\widetilde{\mathbf{P}}$ has been reversed when activating the high-frequency auxiliary mode, exactly like in the measurements of Ref. [29]. Similar to these experiments too, this reversal is partial (i.e., the ratio shown in Fig. 1d never reaches −1) and is transient in nature, since the polarization becomes positive again and approaches $\widetilde{\mathbf{P}}(0)$ at longer times. Note that, in our present case, the full recovery to its initial values of $\widetilde{\mathbf{P}}(t)$ happens at larger times than in Ref. [29], which is likely due to some material specificity.

For instance, the system studied there is LiNbO$_3$ rather than KNbO$_3$, and thus possesses strong oxygen octahedral rotations that have the tendency to interact with polarization[47–50]. On a microscopic level, we found that the dip of the red curve is caused by local dipoles subject to the pulse being aligned along the reverse [$\bar{1}1\bar{1}$] direction or deviating away from the original [11$\bar{1}$] direction.

Furthermore, Ref. [29] also suggested that with an even stronger laser pulse (out of their laser device's reach) a full reversal would have been achieved. However, our numerical simulation (black in Fig. 1d) shows that this is not the case. As a matter of fact, with laser pulses up to 22.27 MV/cm, this full reversal never happened, but rather the ratio shown in Fig. 1d gets close to zero before $\widetilde{\mathbf{P}}(t)$ increases toward $\widetilde{\mathbf{P}}(0)$. One may wonder if this lack of full reversal stems from the fact that only a part of the sample is excited about the results of Fig. 1d. To check such a possibility, Fig. 1e reports the corresponding predictions but when the whole system is subject to the same aforementioned pulse. Note that since we do not expect clamping from a non-excited part of the material, we now allow the homogeneous strains to relax. The polarization can indeed be fully reversed within an R-phase by a field of 12.46 MV/cm magnitude (red curve). However, unlike previously expected[29,38], the full reversal does not happen during the pulse but after. In addition, we numerically find that the polarization can also reach a final state where it points along a direction oblique to [11$\bar{1}$]. For instance, an electric field of E = 8.90 MV/cm gives rise to a final state with half of the material presenting polarization along the rotated [$\bar{1}1\bar{1}$] direction (71° rotation) and with the other half still along the initial [11$\bar{1}$] direction (green curve in Fig. 1e). Similarly, a larger electric field E = 14.24 MV/cm induces a polarization being now along other directions, namely [$\bar{1}\bar{1}\bar{1}$] (109° rotation), [$\bar{1}11$] (71° rotation), and [$\bar{1}\bar{1}\bar{1}$] (180° rotation), the black curve in Fig. 1e, in different parts of the material at the end. This explains why the ratio of Fig. 1e is equal to 0.5 and –0.2 for E = 8.90 MV/cm and E = 14.24 MV/cm, respectively. Details of the evolution of each of the polarization components can be found in Sec. III of SM. The results of Fig. 1e therefore reveal that rotation of polarization in a ferroelectric perovskite, rather than "only" its reversing, has also to be considered and understood, when subjecting a ferroelectric to a high-frequency electric field. Our results so far show that having a three-dimensional ferroelectric like KNbO$_3$ allows a wide range of possibilities for polarization control. Clearly, having a three-dimensional model of the interactions between **P** and **Q,** in this case, is essential. We now focus on simple cases, such as T- and O-phases under pulses, to explore and understand such possibilities.

**Squeezing effect**. Let us thus take a T-phase at 400 K with a polarization lying along the positive $z$-direction, with all the degree of freedoms being allowed to evolve (e.g., **P**, **Q**, inhomogeneous and homogeneous strains), as a simple case. We assume that the whole system is excited by a laser pulse $\mathbf{E}e^{-2\ln 2(\frac{t}{\tau})^2}\cos(2\pi\omega t)$, for which $\tau = 600$ fs, $\omega = 19$ THz and **E** is applied along the $z$-direction with a magnitude of 7.71 MeV/cm, as depicted in Fig. 2a. Such pulse naturally leads to $Q_z$ resonating, as evidenced in Fig. 2b. Within the FWHM (600 fs rather than 200 fs in order to have more oscillations of some components of Q around zero) of the pulse, two striking features emerge and are seen in Fig. 2c: (1) $P_z$ decreases its absolute value to zero, and 2) $P_x$ and $P_y$ adopt non-zero values (with a negative and positive signs, respectively). Consequently, $P_z$ does not reverse itself after going through zero. It rather keeps oscillating around zero, which means that when the polarization is at a zero value the reversal driving force as predicted in Refs. [29,38] vanishes and stops to push $P_z$ toward the opposite direction. After the pulse has occurred, the system chooses an O-phase with polarization along the [$\bar{1}10$]

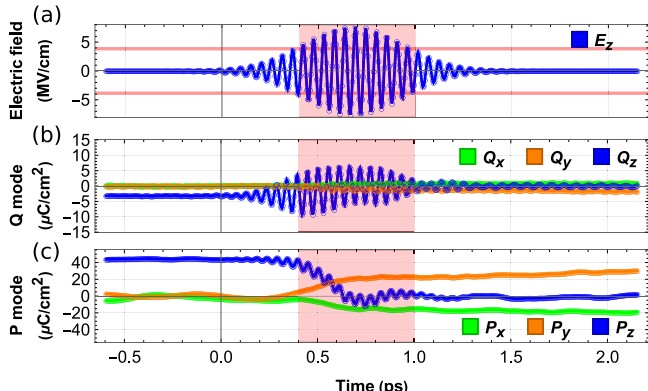

**Fig. 2 Squeezing effect in T-phase.** Temporal behavior of the **Q** (**b**) and **P** (**c**) modes at 400K, as a response to the electric field's pulse shown in **a** and when starting from a T-phase (note that the full-width-half-maximum of the pulse is marked by pink regions).

direction (90° rotation)—via the creation of $x$- and $y$-components and $P_z$ continuing to be around zero. (Such O-phase in some of our simulations will further transform back to another T-phase, such as those with polarization along [$\bar{1}00$] or [010], while in other simulations this O-phase will have a longer lifetime at 400 K.) The applied pulse therefore would prefer to annihilate the component of the polarization that is along the field's direction, in favor of polarization components that are perpendicular to the field. The mid-infrared pulse thus acts like a "squeezer" that "presses" the material along the pulse polarized direction and reduces the magnitude of the component of the polarization that is along the field. To demonstrate that the direct coupling between electric field and **P** mode is not the reason that **P** is squeezed, we ran a numerical experiment in which the degree of freedoms of the **P** mode, inhomogeneous and homogeneous strains are allowed to evolve while the **Q** mode is kept frozen to its equilibrium initial value. In that case, no squeezing effect was observed and $P_z$ was "only" oscillating around its equilibrium value in the initial T-phase instead of being annihilated (see Sec. IV of SM). The activation and response of the **Q** mode to the pulse are therefore required to induce the squeezing effect of **P**.

To understand such a squeezing effect, we need to have a detailed look at Table 1 and realize that, for the case of a T-phase having a z-component of the polarization, the following couplings are relevant: the bi-quadratic

$$U_{22}^{(z)} = \Lambda_2 P_z^2 Q_z^2 \tag{1}$$

$$U_{22}^{(x-z)} = \Lambda_{22}(P_x^2 + P_y^2)Q_z^2 \tag{2}$$

When a mid-infrared laser pulse polarized along z is applied, the high-frequency auxiliary mode $Q_z$ resonates with a large amplitude. The positive $U_{22}^{(z)}$ interaction (see Table 1 for its coefficient $\Lambda_2$) then gives rise to a large positive quadratic term $\Lambda_2 Q_z^2 P_z^2$ felt by $P_z$. Let us understand the consequence of such a large positive term on the polarization, by considering a fourth-order model for the free energy, namely $F = \alpha P_z^4 - \kappa^2 P_z^2 + \Lambda Q_z^2 P_z^2$. The polarization can thus have two minima (hence a double-well potential), that are $P_z^\pm = \pm\sqrt{(\kappa^2 - \Lambda Q_z^2)/(2\alpha)}$ (note that $\alpha$ is always positive to have a bounded energy potential). If $\Lambda$ is positive and $\kappa^2 - \Lambda Q_z^2 > 0$, the equilibrium value of $P_z$ shrinks in magnitude (and the double-well potential becomes shallower) when $Q_z^2$ is growing. Such double well can even transform into a single well with its minimum at $P_z = 0$, when $\kappa^2 - \Lambda Q_z^2$ becomes negative. The wiggly decreasing-in-magnitude of $P_z$ to zero in Fig. 2c within the FWHM of the pulse,

corresponds to the oscillation of $P_z$ within a shrinking double-well potential. Moreover, the growing of $P_x$ and $P_y$ in Fig. 2c within the FWHM of the pulse is favored by the fact that $U_{22}^{(x-z)}$ has a negative coefficient $\Lambda_{22}$ in front of it and becomes stronger when $Q_z^2$ is enlarged through its resonance. Considering that usually the crossing term $\Lambda_{22}$ is much smaller than $\Lambda_2$ (one order of magnitude smaller), the decreasing of the polarization component that parallels the field should be dominant over the increasing of the perpendicular component. Note that the squeezing effect decides along which direction the polarization should be induced but not the sign of this polarization. For instance, the final result at 2 ps of Fig. 2c for the axis for which **P** is parallel could also have been [110], [1̄10], or [1̄1̄0]. Note too that we have also performed various numerical experiments such, as, e.g., switching in and off some interactions in the simulations, that confirm the role of $U_{22}^{(z)}$ and $U_{22}^{(x-z)}$ on the aforementioned effects. It is also realized that the $\Lambda_3$ and $\Lambda_1$ couplings in Table 1 decide the direction of the **Q** mode with respect to that of the **P** mode, details can be found in Sec. IV of SM.

It is worthwhile to emphasize that this squeezing effect is a new phenomenon that originates from the high-frequency auxiliary mode. It should not be accounted as the reduction of the polarization by the thermal effect from the light[51–53] because Fig. 2 clearly shows that though one component ($P_z$) decreased to zero, other components ($P_x$ and $P_y$) condense along with it, while heating will typically result in a reduction of the polarization magnitude overall.

This squeezing effect also allows us to understand the results of partial (transient) and full (permanent) reversal of polarization in Fig. 1d, e for some fields within an R-phase. As a matter of fact, pulses with fields oriented along the [11̄1̄] axis and with the resonant 18 THz frequency induce a strong oscillation of the **Q** mode along such [11̄1̄] axis within the R-phase. During the large oscillation of **Q** (and resulting in large $Q_x^2$, $Q_y^2$ and $Q_z^2$), the **P** mode starts to decrease its magnitude toward zero according to the squeezing mechanism along all three Cartesian directions. After the exit of the 200 fs FWHM pulse, **P** has just reached and oscillated around zero with net momentum toward the reverse [1̄1̄1] direction. If it is a full-excitation case (Fig. 1e, in which all unit cells are subject to the electric field), **P** can continue to grow along the [1̄1̄1] direction and the full reversal of the polarization then occurs. If it is a partial-excitation case (Fig. 1d, in which $10 \times 10 \times 10$ cells within the $12 \times 12 \times 12$ supercells are allowed to be coupled to the electric field), **P** can partially grow along [1̄1̄1] but then quickly reverses its direction and grows back to [111̄], as a result of the interaction with the unexcited part of the material; the transient partial reversal of the polarization thus happens. We also note that Abalmasov[54] proposed that the depolarizing electric field can yield a similar transient partial reversal in thin films. Note that the Supplemental Materials also explain why the polarization can rotate, rather than being reverted, under full-excitation for some fields.

**Deterministic full-reversal strategy**. As can be seen in Fig. 1e, the ferroelectric polarization full reversal is sometimes replaced by a polarization rotation. Consequently, a single laser pulse, such as the one used in the experiment of Ref. [29], will not reverse the polarization of a three-dimensional ferroelectric in a deterministic manner. (Presumably, the reversal will not be deterministic for a one-dimensional compound like $LiNbO_3$ either, unless one implements a way to avoid the back-switching to the original state). However, the understanding of the squeezing effect can enable us to further design a strategy at room temperature to realize a full reversal of the ferroelectric polarization by laser

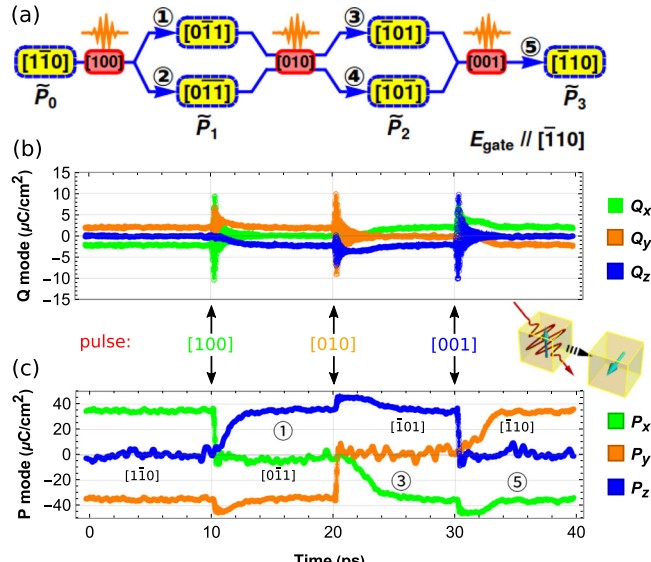

**Fig. 3 Deterministic full-reversal strategy. a** Protocol of the three-pulse reversal process; the yellow background boxes are for the initial and excited polarization phases and the pink background boxes are used to indicate the laser pulses that are along x, y, and z axes; The labels of the phases $\widetilde{\mathbf{P}}$ is defined as **P** + **Q** to indicate the total polarization. The blue arrows are used to indicate the phase transition channels after the laser pulses. Responses of the **Q** (**b**) and **P** modes (**c**) excited by a sequence of three "200fs-18THz" pulses activated at 10, 20, and 30 ps and having their polarized direction along [100], [010], and [011] directions, respectively. For each of these pulses, the magnitude of **E** in $\mathbf{E}e^{-2\ln 2(\frac{t}{\tau})^2}\cos(2\pi\omega t)$ is 6.17 MV/cm.

pulses aided with a tiny dc and constantly applied gate field, which is of high importance for building novel ferroelectric-based devices. Then, we take advantage of the following expected features: (1) that the squeezing effect wants to suppress the component of the polarization that is aligned along the pulse's direction; (2) when forming from a zero value, the components of the polarization will adopt the same sign as those of the applied weak dc field; and (3) the system wishes to stay in an O-phase due to the constant temperature of 300 K (for which O-phases are the equilibrium state). We can thus anticipate, as displayed in Fig. 3a, that (i) a first pulse applied along [100] will lead to a new polarization, $\widetilde{\mathbf{P}}_1$, being now either along [01̄1] (channel ① in Fig. 3a) or [01̄1̄] (channel ② in Fig. 3a); (ii) a second pulse but now applied along [010] either transforms $\widetilde{\mathbf{P}}_1$ being parallel to [01̄1] to $\widetilde{\mathbf{P}}_2$ being along [1̄01] (channel ③ in Fig. 3a) or $\widetilde{\mathbf{P}}_1$ being along [01̄1̄] to $\widetilde{\mathbf{P}}_2$ being parallel to [1̄01̄] (channel ④ in Fig. 3a); and finally (iii) a third pulse, now along [001] should lead to a $\widetilde{\mathbf{P}}_3$ polarization along [1̄10] (channel ⑤ in Fig. 3a) for the two possible $\widetilde{\mathbf{P}}_2$ starting points (channels ③ and ④). If that is the case, we will thus have achieved a fully deterministic (3-pulse process) reversal of the polarization from the initial to final states. It is worth mentioning that the use of several pulses is a common strategy to switch ferroelectric polarization in a deterministic manner[55] and has also been suggested by other theoretical work[56].

In Fig. 3b, c, we present one representative of our many numerical examinations, in order to check our proposed strategy and to explain it in even more detail. An equilibrium O-phase, having $P_x = -P_y > 0$, along with weak $Q_x = -Q_y < 0$, is thus chosen at room temperature (300 K) as the initial state. Three pulses are applied along different directions and at different times, along with a bias dc gate field constantly applied along the

[$\bar{1}10$] direction. The magnitude of this dc field is rather weak, namely 0.0154 MV/cm which is only 3.3% of the minimum dc field needed to reverse the O-phase polarization in our numerical model and about 0.35% of the magnitude of the laser pulse to be applied. As indicated above, the role of this tiny dc field is to bias the $x$ and $y$-components of $P$ to have a negative and positive signs, respectively, when forming, since squeezing effects alone do not guarantee the sign of such components. Practically, a "200 fs-18 THz" laser pulse (i.e., a pulse with $\tau = 200$ fs and frequency $\omega = 18$ THz in $\mathbf{E}e^{-2\ln 2(\frac{t}{\tau})^2}\cos(2\pi\omega t)$) is first applied at 10 ps with E = 6.17 MV/cm and along the [100] direction on this O-phase having a polarization along the [$1\bar{1}0$] direction (note that 18 THz is close enough to the resonant frequency of the auxiliary mode of the O-phase at 300 K). This pulse thus resonates with the auxiliary mode $Q_x$ (see Fig. 3b), and, as expected by the squeezing effect, (i) the component of the polarization along the $x$-direction, $P_x$, vanishes, as shown in Fig. 3c, and (ii) the $y$-axis, along which there is no pulse field, sees its component of polarization, $P_y$, to remain finite and even to further grow in magnitude. Concomitantly, $Q_y$ also slightly increases in its absolute value, because of its $\Lambda_3 P_y^3 Q_y$ and $\Lambda_1 P_y Q_y^3$ couplings with $P_y$. Since the minima of the free energy at 300 K are orthorhombic, while $P_x$ is squeezed to zero, the $P_z$ component develops and takes a positive value in this particular numerical experiment. Note that, beyond the fact that the material will try to adopt an O-phase at this temperature, the squeezing effect of the form $\Lambda_{22} P_z^2 Q_y^2$ (i.e., an equivalent form of Eq. (2), but for $P_z$ and $Q_y$) will also contribute to the development of $P_z$. Note also that, since no dc field is along the $z$-axis, the newly formed $P_z$ can also be negative, which corresponds to the channel ② in Fig. 3a and which we also observed in our other numerical experiments (see Sec. V of SM). We have thus succeeded to make a transition from an O-phase to another O-phase (channel ① at 300 K via a rotation of the polarization by 60°, by activating the auxiliary high-frequency mode at around 10 ps. The second pulse is also a "200 fs-18 THz" one with the same magnitude E = 6.17 MV/cm, but now having a [010] polarized orientation, which then, according again to the squeezing effect, results in the vanishing of the $P_y$ (and also of $Q_y$) in favor of the activation of $P_x$ (and also $Q_x$) that remains finite until 30 ps. The formation of $P_x$ at 20 ps has the same origin as the formation of $P_z$ at 10 ps, but $P_x$ now has to adopt a negative sign because of the small dc field's component along [$\bar{1}00$]. Another O-phase with polarization along [$\bar{1}01$] has therefore been created, corresponding to the channel ③ in Fig. 3a. Similarly, a third "200 fs-18 THz" pulse with the same magnitude E = 6.17 MV/cm is applied at 30 ps, but with a pulse field now oriented along [001]. Consequently, squeezing effects lead to $P_y$ now becoming finite (and positive thanks to the gate field) while $P_z$ vanishes, and $P_x$ remains negative. An O-phase, but now with a polarization fully reversed with respect to the initial O-phase, has thus formed.

The strategy of Fig. 3, therefore, allows us to realize a full reversal of polarization in a deterministic fashion, via three steps each taking advantage of the squeezing effects (which eliminate the polarization component that is parallel to the pulse polarization direction, while inducing the formation of the polarization along the perpendicular directions) combined with the application of a weak dc field that controls the sign of the newly formed component of the polarization. Section V of the SM further provides details on this deterministic control by showing results of the different paths indicated in Fig. 3a, when changing the magnitude of the laser pulses. It is also worth mentioning that a deterministic rotation of the polarization by 60° is obviously possible. For instance, if the weak dc bias field is applied along [001], a single $x$-polarized pulse will deterministically transform

an O-phase with a polarization along [110] phase to another O-phase but with a polarization along [011] at room temperature in KNbO₃.

In conclusion, we developed a method that allows us to model the pulse stimuli on the ferroelectric polarization by resonating an auxiliary high-frequency mode, taking the prototypical ferroelectric material KNbO₃ as a testbed. Our study has revealed and explained how one can obtain a partial transient ferroelectric polarization reversal in SHG experiments, shedding light on the results of Ref. [29]. We also show that, and explain why, a full reversal of the polarization, as well as phase transitions induced by rotation of the polarization, can occur as a response to different polarized laser pulses activating different Cartesian components of the auxiliary mode. We hope the mechanisms and strategies for light-assisted ferroelectric control shown here will stimulate further fundamental research and open new avenues for the design of novel optoferroic devices.

## Methods

**Effective Hamiltonian**. A novel effective Hamiltonian ($H_{eff}$) is developed for such material. It has the following degrees of freedom: vectors related to the ferroelectric soft mode ($P$), high-frequency auxiliary mode ($Q$), and inhomogeneous strain ($\mathbf{u}$) in each 5-atom unit cell, as well as, the homogenous strain ($\eta$). Both $P$ and $Q$ modes are infrared-active modes and can couple to external electric fields (via an energy involving a dot product with such field). Their associated local vectors in the $H_{eff}$ are centered on Nb ions. In the KNbO₃ cubic structure, the zone-center $P$ mode is soft (in the sense that the square of its frequency is negative at 0 K) while the $Q$ mode is also located at the zone center but possesses a high and positive frequency. The details of the mode frequency and eigenvector for both $P$ and $Q$ can be found in Sec. I of SM. The local vectors corresponding to the inhomogeneous strains are technically centered on K ions. The homogeneous strain is defined with respect to cubic symmetry and has six independent components $\eta_i$ in Voigt notation.

The potential energy $U^{tot}$ of $H_{eff}$ has four main contributions:

$$\begin{aligned} U^{tot} = \; & U^{FE}(\{P\}, \{\mathbf{u}\}, \{\eta\}) \\ & + U^{aux}(\{Q\}, \{\mathbf{u}\}, \{\eta\}) \\ & + U^{int}(\{P\}, \{Q\}) \\ & + U^{elastic}(\{\eta\}) \end{aligned} \tag{3}$$

a purely elastic one $U^{elastic}$; a second one related to the ferroelectric soft mode and its interaction with strains $U^{elastic}$, exactly like in Ref. [47] for BaTiO₃; a third one related to the high-frequency auxiliary mode and its interaction with strains $U^{aux}$, which has the same analytical form as the second energy but when replacing $P$ by $Q$; and a fourth one that gathers the direct interactions between the ferroelectric soft mode and high-frequency auxiliary mode $U^{int}$, and which are indicated in Table 1. The details of all these four energies can be found in Sec. II of SM.

Such an effective Hamiltonian is generated with respect to cubic (C) symmetry and can be involved in phase transitions among sub-group ferroelectric structures, such as tetragonal (T), orthorhombic (O), and rhombohedral (R) phases. Note that this new effective Hamiltonian resolves symmetry-broken issues expressed in previous works[29,38,40], and contains for the first time all the symmetry-allowed forms within fourth-order. In addition, for the first time too, the model includes the full spatial degree of freedoms of both $P$ and $Q$, i.e., their Cartesian components along $x$, $y$, and $z$ axes (which are along the three < 001 > pseudo-cubic directions). The existence of these Cartesian components ($P_x$, $P_y$, $P_z$) and ($Q_x$, $Q_y$, $Q_z$) can, e.g., allow the ferroelectric polarization to rotate (and not "only" reverse its direction) when the $Q$ mode is activated. The model parameters of the $H_{eff}$ are first determined from 0 K first-principle calculations but then a few of them are adjusted, in order to better agree with measurements and also to allow the calculations to converge (large coupling coefficients can result in simulations going toward infinite energy). This is why, for instance, the coefficient in front of (the third-order-in-P) $P_x^3 Q_x + P_y^3 Q_y + P_z^3 Q_z$ and (the complex) $P_x^2 P_y Q_y + P_x P_y^2 Q_x + P_x^2 P_z Q_z + P_x P_z^2 Q_x + P_y^2 P_z Q_z + P_y P_z^2 Q_y$ have been reduced from its ab initio value or why the coefficient in the front of $P_x P_y Q_x Q_y + P_y P_z Q_y Q_z + P_z P_x Q_z Q_x$ has been modified (to give correct A and E mode splitting hierarchy, see Supplementary Fig. 2). Note also that determining such coupling coefficients from first principles is not a unique and easy procedure and thus large error bars can be assigned to these coefficients. It is thus not surprising that they then require to be adjusted to allow for a better comparison with observations.

Moreover, the kinetic energy $K^{tot}$ of the $H_{eff}$ contains three parts written with respect to the velocity $\mathbf{v}_p$, $\mathbf{v}_q$, and $\mathbf{v}_u$ that correspond to the order parameters, $P$, $Q$, and $\mathbf{u}$:

$$K^{tot} = \sum_i^N \frac{1}{2} M_p v_{p,i}^2 + \frac{1}{2} M_q v_{q,i}^2 + \frac{1}{2} M_u v_{u,i}^2 \tag{4}$$

where the effective masses are defined as $M_p$, $M_q$, and $M_u$, respectively. Their values are fitted to produce the TO mode frequencies as calculated from the DFT phonon frequencies of the rhombohedral phase at zero K[57], and are listed in Supplementary Table I.

**Dynamic simulations**. We employed MC and MD algorithms on a $12 \times 12 \times 12$ supercells that contains 8640 atoms. More specifically, we used parallel tempering[58,59] (PT) MC and Nosé–Hoover thermal state[60–62] MD simulations implemented in LINVARIANT[63]. The same Nosé mass for **P**, **Q**, and inhomogeneous strains of 10,000 a.u. is used. The Nosé mass for the homogenous strains is 1 a.u. in all the simulations. Periodic boundary conditions were adopted. For each temperature, 150,000 PTMC sweeps are firstly performed, with the first 100,000 steps as thermalization and the subsequent 50,000 steps to compute the phase diagram; then MD simulations are initialized with the MC outputs, and 500,000 thermalization steps are performed before the statistical evaluations; the time interval of 0.1 fs is used in the MD simulations.

## Data availability
The authors declare that the coefficient data of the effective Hamiltonian are available within the paper and its Supplementary material. The data that support the findings of this study are available from the corresponding author upon reasonable request.

## Code availability
LINVARIANT[63] code is used to generate and solve the effective Hamiltonian.

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

## Acknowledgements

The work is supported by ONR under Grant No. N00014-17-1-2818 (P.C. and L.B.), the Vannevar Bush Faculty Fellowship (VBFF) grant no. N00014-20-1-2834 from the Department of Defense (H.J.Z and L.B.), and the ARO Grant No. W911NF-21-1-0113. (L.B.). C.P. thanks the support from a public grant overseen by the French National Research Agency (ANR) as part of the "Investissements d'Avenir" program (Labex NanoSaclay, reference: ANR-10-LABX-0035) and ANR grant THz-MUFINS (Grant No. ANR-21-CE42-0030). J.Í. is funded by the Luxembourg National Research Fund through Grant FNR/C18/MS/12705883/REFOX. The simulations of effective Hamiltonian and density functional theory were done using the Arkansas High Performance Computing Center.

## Author contributions

L.B. and J.Í. conceived the work; P.C., C.P., and L.B. implemented the effective Hamiltonian and performed numerical simulations; P.C., C.P., H.J.Z., and L.B. carried out the analysis and explanation of the data; all authors participated in the discussion and preparation of this work.

## Competing interests

The authors declare no competing interests.
