## [Peer Review File · Nature Communications]

Reviewers' Comments:

Reviewer #1:

Remarks to the Author:

The manuscript 'Deterministic control of ferroelectric polarization by ultrafast laser pulses' by Cheng et al. deals with the dynamical response of the ferroelectric polarization (P) to the selective excitation of high frequency phonon modes (Q) in perovskite materials. The present study applies Monte Carlo simulations with a 12x12x12 supercell and an effective Hamiltonian, which includes the coupling terms. The quartic coupling between Q and P results in the transformation of the double well potential of P into one with a single minimum at P=0 as the amplitude of Q increases. Interestingly, only the component of P aligned with the polarization of the phonon mode is reduced substantially, resulting in a rotation of the polarization into the plane perpendicular to Q. Based on these findings, the authors present a recipe for switching the initial polarization in a three step process. Three consecutive electromagnetic pulses with different polarizations are used to subsequently rotate the polarization until it reaches the final state. This idea is experimentally hard to realize. While the electric field, pulse duration and frequency of the pump pulses are easy to reach, it is not trivial to excite the same volume of the sample with electromagnetic fields polarized along all three crystallographic directions. As the authors note, the concept can also be applied to the rotation of the polarization component projected onto a two dimensional plane, which would already be very interesting and relevant.

The authors further validated their model by comparing the predicted dynamics to the experimental results reported on LiNbO₃, where the polarization has been found to invert partially, but only for a short time. The model reproduces the experiment qualitatively if the polarization of the unit cells on the edges of the supercell are frozen. These results indicate that stable polarization reversal on ultrafast timescales is not possible as long as only part of a bulk sample is excited or domains are formed.

There have been several theoretical studies, which investigate the effect of coherent phonon excitation on other coupled modes in ferroelectrics, magnetic materials, superconductors and others. Many of these studies also employ effective Hamiltonians with coupling constants derived from frozen phonon density functional theory calculations. However, to my knowledge, none of these studies employed Monte Carlo simulations to investigate the response of the system in all three dimensions. Furthermore, the model is not limited to ferroelectrics and the energy potential squeezing described here can play a role also in other materials showing similar quartic coupling terms. As such, the manuscript is of general interest, which justifies publication in Nature communications.

However, I would like to discuss some points before I can recommend publication.

1. In the present paper, an electromagnetic pulse with 25 THz center frequency was chosen for the simulations. 25 THz corresponds to the highest energy phonon mode longitudinal optical (LO) frequency (i.e. the zero crossing of $\text{Re}(\epsilon)$) of KNbO₃, while the TO frequency (peak of $\text{Im}(\epsilon)$) is 15.6 THz (M D Fontana et al 1984 J. Phys. C: Solid State Phys. 17 483). In linear response, the electromagnetic field is only absorbed at the TO frequency, not at the LO frequency. This raises several questions:

1.a. In the supplemental material, the authors present the ϵ_{11} element of the complex dielectric tensor but it is not clear to me, what is actually shown ($\text{Im}(\epsilon)$, $\text{Im}(1/\epsilon)$, $\text{Re}(\epsilon)$, $\text{Abs}(\epsilon)$). Also, why is an arbitrary unit used and why is there a peak at 25 THz as opposed to 15.6 THz for the Q mode?

1.b. If 25 THz corresponds to the LO frequency, how do the dynamics change if the frequency is tuned to the TO frequency?

1.c. What is actually driving the squeezing of the ferroelectric mode potential, specific atomic motions or an induced polarization including not only ionic but also electronic polarization?

1.d. Related to the questions above: The authors write that the linear response to the electric field is given by $E * Q / (\omega^2 - \omega_i^2)$, where ω_i is the Eigenfrequency of the oscillator. This Eigenfrequency is normally the TO frequency, not the LO frequency of 25 THz. The TO-LO splitting is the result of the dielectric properties of the material and the oscillator strength.

2. The simulation was done on a 12 x 12 x 12 supercell. I appreciate that larger clusters require long computation time but I am missing a discussion about which physical effects can be investigated with a cluster of this size, and which are inaccessible. An edge length of 5nm is still very small for electrostatic forces. Also, were the results compared to simulations on smaller clusters? The freezing of the polarization at the edges of the 5nm x 5nm x 5nm cluster was required to simulate the experimental results on LiNbO₃, where the excited sample volume was on the order of hundreds of micron in area and a few micron in depth. Can the authors comment, why the model can still be compared to the experiment?

3. While the description of the methods and the recipe for polarization rotation is clear, the general style of the manuscript has to be improved.

Some points I came across:

Line 29: e.g. can be omitted

Line 51: the authors last name should be used: Subedi

117: experiments instead of experiment

134: ',' instead of 'and'

181: I don't understand the expression 'pulse starting at $t=0$ ps and ending around $t=5$ ps.' - is the pulse duration not 250 fs?

182-184: 'Note that, in simulations with this type of effective models, the applied electric fields are typically predicted to be 20 times larger than the experimental ones⁴²'
What is meant by 'the predicted electric fields' and why are they 20 times larger?

291: 'desires to vanish' is not a good formulation here

339: To which time after the excitation does 'final result' refer to?

370: It is not clearly explained that full-excitation refers to the excitation of all unit cells in the cluster, while partial excitation refers the polarization being frozen at the edges of the cluster. Maybe this can be defined more clearly.

Reviewer #2:

Remarks to the Author:

The article addresses a rather acute issue of proving a possibility of a permanent reversal of ferroelectric polarization by an electromagnetic pulse and, in particular, by a resonant electromagnetic pulse.

The urgency of this problem is caused by the possibility of using ferroelectrics in devices for all optical switching (AOS) ("optical" in this context is commonly understood rather broadly: from real optics to the THz range), the new exciting physics of the ultrashort optical pulses impact over order parameters, and, finally, "envy" of magnetism, in which it was possible not only to switch magnetization in the AOS mode, but also to adequately describe the ongoing processes (magnetic issues as well as some others are reviewed quite well by the authors of the paper).

In their approach, the authors use classical Hamiltonian for ferroelectric perovskite of Zhong (Ref. 37) and the idea firstly suggested in [28] of the resonant excitation not directly of the soft mode but of a higher-frequency polar-active mode and nonlinear interaction between them. From point of view of theory, this is a big step forward: the model calculations are described by the authors in great detail and, on the whole, look convincing.

In general, the article satisfies the requirements of the Journal in terms of originality, novelty and

importance for science and its applications, but for publication it requires revision on the listed issues:

1. General

1.1. Transient ferroelectric reversal by a broadband THz pulse (with the soft mode frequency falling within the excitation spectra) was observed experimentally by optical probe as well as by XRD [ArXiv ID 1602.05435 (2016), 10.1103/PhysRevLett.118.107602, 10.1038/s41598-017-00704-9, 10.1002/pssr.202000460]. It seems necessary to refer to these works.

1.2. Are there any proof that direct resonant excitation of the soft mode deterministically cannot provide permanent ferroelectric polarization reversal?

1.3. The choice of the crystal is not justified.

1.4. The term "squeezed" seems to be confusing.

2. It is necessary to give explanation regarding the following parameters:

2.1. THz pulse width: why for modeling rhombohedral phase the pulse width is 200 fs, while for tetragonal phase it is 600 fs? What is the dependence of the discussed effects on the pulse width?

2.2. Soft mode and other frequencies: All the important frequencies are the result of DFT calculations. It is necessary to compare these values with the results of calculations by other authors and, more importantly, with the available experimental results (See, for example, Ref. [10.1103/PhysRevB.54.11161]).

3. "Deterministic full-reversal strategy".

3.1. This is the most muddy part. A lot of details are presented (and some of them are repeated), behind which the main idea is lost. I propose to clearly highlight the main elements of the strategy, number them, and send the rest of the text to SM.

3.2. Is it really possible to recreate in an experiment all the conditions required in the theory? Is it really necessary to take such a complex crystal for this? Is the theory only suitable for this crystal? Is it possible to estimate the probability of success of this strategy application for different ferroelectric materials?

3.3. The use of several pulses is a common strategy to switch ferroelectric polarization in deterministic manner [10.1063/1.4974953]. Moreover, pulse shaping as combination of several THz pulse for efficient and ultrafast polarization switching was suggested in [10.1103/PhysRevLett.102.247603], it is necessary to refer.

3.4. The domain structure, which can be quite complex and influence strongly on the polarization switching, is not considered at all. Couldn't it lead to the collapse of the entire model?

3.5. Depolarization field is not taken into account and even is not mentioned. This can also hinder success [10.1103/PhysRevB.101.014102].

4. Minor remarks

4.1. Line 51 "... proposed by Alaska and his coworkers..." should be changed by "... proposed by A. Subedi and his coworkers..." (Subedi is the last name).

4.2. Fig.2 X-axis, units are absent (time scale), please add.

4.3. Supplementary Fig.1. a) plus in the circle - explain the meaning in the legend

Response Letter

Reviewer #1:

1. Remarks to the Author: *The manuscript 'Deterministic control of ferroelectric polarization by ultrafast laser pulses' by Chen et al. deals with the dynamical response of the ferroelectric polarization (P) to the selective excitation of high frequency phonon modes (Q) in perovskite materials. The present study applies Monte Carlo simulations with a $12 \times 12 \times 12$ supercell and an effective Hamiltonian, which includes the coupling terms. The quartic coupling between Q and P results in the transformation of the double well potential of P into one with a single minimum at $P=0$ as the amplitude of Q increases. Interestingly, only the component of P aligned with the polarization of the phonon mode is reduced substantially, resulting in a rotation of the polarization into the plane perpendicular to Q . Based on these findings, the authors present a recipe for switching the initial polarization in a three step process. Three consecutive electromagnetic pulses with different polarizations are used to subsequently rotate the polarization until it reaches the final state. This idea is experimentally hard to realize. While the electric field, pulse duration and frequency of the pump pulses are easy to reach, it is not trivial to excite the same volume of the sample with electromagnetic fields polarized along all three crystallographic directions. As the authors note, the concept can also be applied to the rotation of the polarization component projected onto a two dimensional plane, which would already be very interesting and relevant.*

The authors further validated their model by comparing the predicted dynamics to the experimental results reported on LiNbO_3 , where the polarization has been found to invert partially, but only for a short time. The model reproduces the experiment qualitatively if the polarization of the unit cells on the

edges of the supercell are frozen. These results indicate that stable polarization reversal on ultrafast timescales is not possible as long as only part of a bulk sample is excited or domains are formed.

There have been several theoretical studies, which investigate the effect of coherent phonon excitation on other coupled modes in ferroelectrics, magnetic materials, superconductors and others. Many of these studies also employ effective Hamiltonians with coupling constants derived from frozen phonon density functional theory calculations. However, to my knowledge, none of these studies employed Monte Carlo simulations to investigate the response of the system in all three dimensions. Furthermore, the model is not limited to ferroelectrics and the energy potential squeezing described here can play a role also in other materials showing similar quartic coupling terms. As such, the manuscript is of general interest, which justifies publication in *Nature communications*.

However, I would like to discuss some points before I can recommend publication.

Answer: We thank this Reviewer for the nice evaluation and detailed comments of our manuscript, which are valuable for us to further improve our manuscript. We also hope that our work will encourage experimentalists to pursue further developments (e.g., to apply three consecutive pulses with different polarizations) in order to confirm and use our predicted effects in devices

2. Remarks to the Author: **1.** *In the present paper, an electromagnetic pulse with 25 THz center frequency was chosen for the simulations. 25 THz corresponds to the highest energy phonon mode longitudinal optical (LO) frequency (i.e. the zero crossing of $\text{Re}(e)$) of KNbO_3 , while the TO frequency (peak of $\text{Im}(e)$) is 15.6 THz (M D Fontana et al 1984 *J. Phys. C: Solid State Phys.* 17 483). In linear response, the electromagnetic field is only absorbed at the TO frequency, not at the LO frequency. This raises several questions:*

Answer: We believe we may have led Reviewer #1 to a misunderstanding. In short: We have both TO and LO modes in our simulations, but only the zone-center modes corresponding to a TO phonon branch couple to the homogeneous electric field applied.

Let us elaborate on this. Note that our model is defined in terms of local variables (e.g., local dipoles at each 5-atom site), and that such local variables can fluctuate with arbitrary modulation – implying that our simulations include both TO and LO bands naturally. Note also that when we apply a homogeneous electric field, such a field can only couple to homogeneous modes contributing to the macroscopic polarization, i.e., with modes at the center of the Brillouin zone. At the zone center, there is no distinction between LO and TO modes, as there is no modulation for $q=0$. However, it is well-known that the optical polar modes at the zone center lie in the TO band, which is continuous at $q=0$. In contrast, the corresponding LO band displays a discontinuity (due to long-range repulsive electrostatic forces that appear for $q \rightarrow 0$) associated to the well-known LO-TO splitting. Hence, in conclusion, the homogeneous electric fields applied in our simulations couple to zone-center modes that we can call “TO” (as they are the limit of the TO band for $q \rightarrow 0$), but never “LO”.

In lines 118~124, we thus clarify that “Note also that the homogeneous electric fields applied in our simulations couple to zone-center modes that we can call “TO” (as they are the limit of the TO bands for $q \rightarrow 0$, with q being reciprocal vectors within the first Brillouin zone), but not “LO” (that involve a discontinuity that appear for $q \rightarrow 0$, due to long-range repulsive electrostatic forces)”.

Moreover, our previous choices of the effective masses for these two TO modes (P and Q) gave inaccurate frequencies, as compared to the experiment mentioned by the Reviewer. For instance, our previous model gave values of about 15 and 25 THz for the A-type TO modes of P and Q modes, while experiments yield about 8 and 18 THz, respectively, in the rhombohedral phase of KNbO_3 . We have now corrected these quantitative discrepancies by varying the effective masses, and the predicted TO frequencies are now about 8.1 and 18.0 THz for the P and Q modes (A type symmetry), respectively, at the temperature of 240 K in the rhombohedral phase of KNbO_3 – which is in excellent agreement with the aforementioned experiments. Note also that we not only get the A type mode at the correct frequency but also a splitting between A type and E type modes that is comparable to experiments by varying two of our model parameters (namely, Λ_{211} and Λ_{1111} , by fitting them to phonons in the rhombohedral phase rather than the cubic phase (as we did before). For instance, as the reviewer pointed out the E-type TO mode for Q mode is at 15.6 THz in experiment, and our value for this frequency is 16 THz at 240 K with our new parameters.

With those aforementioned changes of model parameters, we re-did all the simulations. For instance, the laser frequencies are now centered around 18 THz for both rhombohedral and orthorhombic phase, and 19 THz for tetragonal phase respectively in the revision. Figures are updated and some of the electric field magnitudes are changed accordingly. Our new results are qualitatively the same as before (they, e.g., confirm the concept of squeezing effect which thus further validate the robustness of our results).

3. Remarks to the Author: 1.a. *In the supplemental material, the authors present the e_{11} element of the complex dielectric tensor but it is not clear to me, what is actually shown ($\text{Im}(e)$, $\text{Im}(1/e)$, $\text{Real}(e)$, $\text{Abs}(e)$). Also, why is an arbitrary unit used and why is there a peak at 25 THz as opposed to 15.6 THz for the Q mode?*

Answer: We also thank the reviewer for this comment. As now indicated in fig.2 of the supplementary materials, e_{11} is $\text{Im}(e)$ and the right unit is used for this element.

4. Remarks to the Author: 1.b. *If 25 THz corresponds to the LO frequency, how do the dynamics change if the frequency is tuned to the TO frequency?*

Answer: Please see our answer to point 2 of this Reviewer.

5. Remarks to the Author: 1.c. *What is actually driving the squeezing of the ferroelectric mode potential, specific atomic motions or an induced polarization including not only ionic but also electronic polarization?*

Answer: As now explicitly stated in the revised version of the manuscript, our effective Hamiltonian does not include electronic degrees of freedom but rather takes into account ionic displacements associated with the Q and P modes. Consequently, the squeezing of the ferroelectric mode at THz frequencies does originate from specific atomic motions. Note also that the electronic excitation, if any, is expected to occur at much higher frequencies (namely, around hundreds of THz) than those studied here.

To address this comment, we have now added in lines 112~114 “Such effective Hamiltonian does not explicitly include electronic degrees of freedom but rather takes into account ionic displacements associated with the Q and P modes.”

6. Remarks to the Author: 1.d. *Related to the questions above: The authors write that the linear response to the electric field is given by $E * Q / (\omega^2 - \omega_i^2)$, where ω_i is the Eigenfrequency of the oscillator. This Eigenfrequency is normally the TO frequency, not the LO frequency of 25 THz. The TO-LO splitting is the result of the dielectric properties of the material and the oscillator strength.*

Answer: Please see our answer to point 2 of this Reviewer. Indeed, the eigenfrequencies used in the linear response are TO modes (for the P and Q modes).

7. Remarks to the Author: 2. *The simulation was done on a 12 x 12 x 12 supercell. I appreciate that larger clusters require long computation time but I am missing a discussion about which physical effects can be investigated with a cluster of this size, and which are inaccessible. An edge length of 5nm is still very small for electrostatic forces. Also, were the results compared to simulations on smaller clusters? The freezing of the polarization at the edges of the 5nm x 5nm x 5nm cluster was required to simulate the experimental results on LiNbO₃, where the excited sample volume was on the order of hundreds of micron in area and a few micron in depth. Can the authors comment, why the model can still be compared to the experiment?*

Answer: Indeed, a 12x12x12 supercell is still small, as compared to the real sample having microns in size, but it is rather large for atomistic simulations. Moreover, we numerically found that 12x12x12 supercells, unlike 10x10x10 supercells, are large enough to better describe the (experimentally-known) first-order phase transitions of KNbO₃. That is why we are confident that a 12x12x12 supercell can qualitatively describe well some phenomena in KNbO₃. Moreover, we tried to understand the “transient” polarization reversal by both analytical analysis of the effective Hamiltonian results and many numerical experiments. Only partial excitation in our numerical experiment is able to produce such “transient” feature. Such transient nature can be understood from the physical picture that emerges from our theory: illumination does not provide a driving force for the polarization to go from P_{initial} to -P_{initial}. Hence, when we turn off the light and relax back to equilibrium, the most probable scenario is for the P_{initial} phase to nucleate at the boundary with the “dark” region (frozen edge in the numerical simulation) and quickly expand to the whole volume. Thus, the recovery of the homogeneous P_{initial} state can be expected, although, likely, the larger the illuminated region, the more difficult to get P_{initial} in 100% of the volume.

Accordingly, in lines 126~129, we have added that “(We numerically found that using 10x10x10 or smaller supercells are not large enough to reproduce the first-order characters of the phase transitions in KNbO₃.)”

8. Remarks to the Author: 3. *While the description of the methods and the recipe for polarization rotation is clear, the general style of the manuscript has to be improved.*

Some points I came accross:

Line 29: e.g. can be ommitted

Answer: Thanks. The manuscript is corrected accordingly.

Line 51: the authors last name should be used: Subedi

Answer: Thanks. The manuscript is corrected accordingly.

117: experiments instead of experiment

Answer: Thanks. The manuscript is corrected accordingly.

134: ',' instead of 'and'

Answer: Thanks. The manuscript is corrected accordingly.

181: I don't understand the expression 'pulse starting at t=0 ps and ending around t=5 ps.' - is the pulse duration not 250 fs?

Answer: Thanks. This sentence was indeed problematic. The expression “*ending around t=5 ps*” was supposed to mean that the whole simulation (rather than the single pulse) ended at 5 ps. It is now changed to “*with pulse starting at t=0 ps*” in line 201.

182-184: 'Note that, in simulations with this type of effective models, the applied electric fields are typically predicted to be 20 times larger than the experimental ones⁴²'

What is meant by 'the predicted electric fields' and why are they 20 times larger?

Answer: According to our experience of working with effective Hamiltonians and comparison with experiment (see, e.g., Xu et al, Nat. Commun. 8, 15682 (2017), when comparing P-E loops in BiFeO₃), the electric field amplitude used in experiment to switch the polarization seems to be 20 times smaller than the corresponding computational one. The most likely reason for that is that, in real ferroelectric materials, defects and nucleation of disordered ferroelectric domains are unavoidable and the electric field applied to switch the ferroelectric polarization of defect-free monodomains should be larger, which is related to the so-called Landauer paradox.

291: 'desires to vanish' is not a good formulation here

Answer: We now change it to “*would prefer to be annihilated*” in line 314.

339: To which time after the excitation does 'final result' refer to?

Answer: To address this question, we now write in line 364 “the final result at 2 ps of fig. 2(c) for the axis for which P is parallel to could also have been [110], [1-10], or [-1-10].”

370: It is not clearly explained that full-excitation refers to the excitation of all unit cells in the cluster, while partial excitation refers the polarization being frozen at the edges of the cluster. Maybe this can be defined more clearly.

Answer: To address this question, we now write in line 397 “If it is a full-excitation case (fig. 1 (e), in which all unit cells are subject to the electric field)” and in line 401 “If it is a partial-excitation case (fig. 1 (d), in which 10x10x10 cells within the 12x12x12 supercell are allowed to be coupled to the electric field)”

Reviewer #2:

Remarks to the Author: *The article addresses a rather acute issue of proving a possibility of a permanent reversal of ferroelectric polarization by an electromagnetic pulse and, in particular, by a resonant electromagnetic pulse.*

The urgency of this problem is caused by the possibility of using ferroelectrics in devices for all optical switching (AOS) (“optical” in this context is commonly understood rather broadly: from real optics to the THz range), the new exciting physics of the ultrashort optical pulses impact over order parameters, and, finally, “envy” of magnetism, in which it was possible not only to switch magnetization in the AOS mode, but also to adequately describe the ongoing processes (magnetic issues as well as some others are reviewed quite well by the authors of the paper).

In their approach, the authors use classical Hamiltonian for ferroelectric perovskite of Zhong (Ref. 37) and the idea firstly suggested in [28] of the resonant excitation not directly of the soft mode but of a higher-frequency polar-active mode and nonlinear interaction between them. From point of view of theory, this is a big step forward: the model calculations are described by the authors in great detail and, on the whole, look convincing.

In general, the article satisfies the requirements of the Journal in terms of originality, novelty and importance for science and its applications, but for publication it requires revision on the listed issues:

Answer: We appreciate the positive feedback and nice summary of our work.

1. General

Remarks to the Author: 1.1. Transient ferroelectric reversal by a broadband THz pulse (with the soft mode frequency falling within the excitation spectra) was observed experimentally by optical probe as well as by XRD [ArXiv ID 1602.05435 (2016), 10.1103/PhysRevLett.118.107602, 10.1038/s41598-017-00704-9, 10.1002/pssr.202000460]. It seems necessary to refer to these works.

Answer: Absolutely. These references have thus been added in:

line 36: “Light-induced switching of ferroelectric polarization [25–34] is also among these most important achievements”

and at

line 58: “measurements “only” reported a transient reversal [26,29]”

Remarks to the Author: 1.2. *Are there any proof that direct resonant excitation of the soft mode deterministically cannot provide permanent ferroelectric polarization reversal?*

Answer: As shown in the figures below, a direct resonant excitation of the soft mode at 8.8 THz does not allow a deterministic reversal of the ferroelectric polarization in the O-phase at 300K when the applied electric field (with Gaussian-type of laser pulse) is as large as the one for high-frequency excitation. This can also be understood by the fact that light does not naturally break the symmetry between +P and -P. Hence, there is no reason to expect a deterministic reversal of P, in our minds.

Interestingly a small squeezing effect seems to still exist, as can be seen that P_x and P_y components decrease their magnitudes almost to zeros at 0.8 ps, while Q_x and Q_y have relatively large oscillation during the pulse. These latter oscillations can be understood by realizing that the Q mode has also resonant frequencies around 8.8 THz (in addition to its main frequencies around 18THz) because of its coupling with the P mode –as revealed by the peaks of the responses shown below both for the P and Q modes. These oscillations of Q can generate squeezing effects for P.

One should also note that the used width of the laser pulse, namely 200 fs, is short, as compared with the time associated with frequency of the soft-mode: the 200 fs laser only contains two cycles of AC field when exciting the soft mode at low frequency (8.8 THz corresponds to 113.6 fs). Consequently, there are few oscillations of P_x and P_y around zero, which prevent the polarization reversal at later times (i.e. after the pulse has vanished). It is thus preferable to excite the high-frequency mode (i.e., of the Q mode) when using short laser pulses (which have less energy cost), in order to induce a reversal of P.

Note also that previous studies, such as the references mentioned in comments 1.1, used a different type of electric pulses (namely, mostly asymmetric) and realized soft mode reversal (with the soft mode frequency falling within the excitation spectra), which is different from our work.

Remarks to the Author: 1.3. *The choice of the crystal is not justified.*

Answer: KNbO_3 is a prototypical ferroelectric with different phases, such as rhombohedral, orthorhombic, tetragonal and cubic, which allows us to explore not only the light-induced polarization reversal but also phase transitions associated with change of polarization's direction. In other words, using a rhombohedral ferroelectric, with the whole sequence of transitions, allows us (1) to have easier polarization rotation (and reversal) and (2) to explore further possibilities for control. A much richer playground than 1-D ferroelectric (LiNbO_3) or tetragonal ferroelectric (PbTiO_3 that has only a single transition from cubic with no polarization to tetragonal with a polarization along $\langle 001 \rangle$) is thus achieved.

Furthermore, a previous effective Hamiltonian (only considering soft mode) exists for this system, which allows us to compare our results with those of this previous Heff but also to demonstrate the effect of the high-frequency mode and its coupling with soft mode on physical properties.

We have thus added in lines 98~105 “It allows us to explore different ferroelectric phases with different polarization's directions at different temperature, which leads to a richer playground than one-dimensional ferroelectrics (e.g. LiNbO_3) or tetragonal ferroelectrics (e.g., PbTiO_3 that has only a single transition from cubic with no polarization to tetragonal with a polarization along $\langle 001 \rangle$).”

Remarks to the Author: 1.4. *The term “squeezed” seems to be confusing.*

Answer: We are not completely sure why this term is confusing. We tried to explain it even more when first defining it in lines 319~321 “The mid-infrared pulse thus acts like a “squeezer” that “presses” the material along the pulse polarized direction and reduces the magnitude of the component of the polarization that is along the field.”

2. It is necessary to give explanation regarding the following parameters:

Remarks to the Author: 2.1. THz pulse width: why for modeling rhombohedral phase the pulse width is 200 fs, while for tetragonal phase it is 600 fs? What is the dependence of the discussed effects on the pulse width?

Answer: This is an excellent question. We used 200 fs for the rhombohedral phase in order to better compare with the experiment done in (Mankowsky et al Phys. Rev. Lett. 118, 197601, 2017). On the other hand, we used a wider pulse of 600 fs, when revealing and discussing the “squeezing effect” in the tetragonal phase, because we want to sustain the oscillations of some components of Q around zero long enough to see the polarization stays in a squeezed state instead of going towards the reversed direction (i.e., being fully reversed). The corresponding words in the revised main text reflect such fact ins line 302~306: “It rather keeps oscillating around zero, which means that when the polarization is at a zero value the reversal driving force as predicted in Refs. [28, 34] vanishes and stops to push Pz towards the opposite direction.”

To make it clear we have added in line 296 “600 fs rather than 200 fs in order to have more oscillations of some components of Q around zero”.

Remarks to the Author: 2.2. Soft mode and other frequencies: All the important frequencies are the result of DFT calculations. It is necessary to compare these values with the results of calculations by other authors and, more importantly, with the available experimental results (See, for example, Ref. [10.1103/PhysRevB.54.11161]).

Answer: This is an excellent point as well. We have updated the dielectric figure (fig.2 in supplementary materials, also attached here), in which the comparison between numerical results and experiments is made for the different resonant frequencies of the P and Q modes for different temperatures. Please also see our answer to point 2 of Reviewer #1, where we explained how we corrected the calculated frequencies.

The black vertical lines are to indicate the experimental values from Ref. [10.1103/PhysRevB.54.11161] and Ref. [J. Phys. C: Solid State Phys. 17 483

(1984)]. As can be seen that the calculated peaks for resonant frequencies are in good agreement with the experiment data of rhombohedral and orthorhombic phases; The deviation of numerical results from the experiment in tetragonal and cubic phases is within 1 THz.

3. “Deterministic full-reversal strategy”.

Remarks to the Author: 3.1. This is the most muddy part. A lot of details are presented (and some of them are repeated), behind which the main idea is lost. I propose to clearly highlight the main elements of the strategy, number them, and send the rest of the text to SM.

Answer: We hope that the schematic flow, fig 3 (a), explains the strategy step by step as the reviewer suggested. We provide detailed explanations in the text to appreciate and help the reader to fully understand this strategy.

Remarks to the Author: 3.2. *Is it really possible to recreate in an experiment all the conditions required in the theory? Is it really necessary to take such a complex crystal for this? Is the theory only suitable for this crystal? Is it possible to estimate the probability of success of this strategy application for different ferroelectric materials?*

Answer: Please first note that these types of experiments have been carried before [e.g. in Cavalleri's group], and we show in Fig. 1c that our results are consistent with results obtained on a different material [LiNbO₃ in the Cavalleri's group]. This highlights already some degree of generality/universality of our results. Secondly, based on symmetry arguments, the couplings involved in this material are general to all perovskite oxides, which should make our conclusions quite broad. Regarding the choice of the system, please see our answer to comment 1.3 of Reviewer #2.

Remarks to the Author: 3.3. *The use of several pulses is a common strategy to switch ferroelectric polarization in deterministic manner [10.1063/1.4974953]. Moreover, pulse shaping as combination of several THz pulse for efficient and ultrafast polarization switching was suggested in [10.1103/PhysRevLett.102.247603], it is necessary to refer.*

Answer: This is a very valid point as well. We thus now quote these references in “It is worth to mention that the use of several pulses is a common strategy to switch ferroelectric polarization in deterministic manner [54] and has also been suggested by another theoretical work [55]” in lines 448-451.

Remarks to the Author: 3.4. *The domain structure, which can be quite complex and influence strongly on the polarization switching, is not considered at all. Couldn't it lead to the collapse of the entire model?*

Answer: Please note that we do observe the formation of multi-domains (see, e.g., the green and black curves in fig1. (e)). Consequently, our model can incorporate the effect of multidomains if the considered system wishes to adopt such multidomains. In the deterministic simulations, the bias gate field is helpful to reduce the creation of multidomains. Thus our 3-pulse and small-bias strategy should take us to the desired destination no matter what the initial point is. As a matter of fact, one can start from a different polarized state, apply the same pulse sequence and then end up in the same final state.

Remarks to the Author: 3.5. *Depolarization field is not taken into account and even is not mentioned. This can also hinder success [10.1103/PhysRevB.101.014102].*

Answer: Indeed, we do not have depolarizing field coming from surfaces because we studied a bulk and not thin films. One reason we choose bulk is because the transient polarization reversal was observed in a 5 mm thick LiNbO₃ crystal (As now mentioned in our revised manuscript in line 200). The depolarization field can have a significant role when investigating thin films, as discussed in the paper *10.1103/PhysRevB.101.014102* that the Reviewer mentioned. This will be studied in future works, in order to separate different effects (e.g., squeezing effects *versus* depolarizing-field effects). We have cited and acknowledged this important work in the revision in lines 406–409 “We also note that Abalmasov [54] proposed that the depolarizing electric field can yield a similar transient partial reversal in thin films.”

4. Minor remarks

Remarks to the Author: 4.1. Line 51 “.... proposed by Alaska and his coworkers...” should be changed by “.... proposed by A. Subedi and his coworkers...” (Subedi is the last name).

Answer: Thanks. The manuscript is corrected accordingly.

Remarks to the Author: 4.2. *Fig.2 X-axis, units are absent (time scale), please add.*

Answer: Thanks. The manuscript is corrected accordingly.

Remarks to the Author: 4.3. *Supplementary Fig.1. a) plus in the circle - explain the meaning in the legend*

Answer: Thanks. The manuscript is corrected accordingly.

Reviewers' Comments:

Reviewer #1:

Remarks to the Author:

The authors have addressed my questions and remarks and the paper has improved considerably. Adjusting the parameters of the simulations to reproduce the experimental values prevents confusion and also validates the used model.

I recommend publication of the manuscript in nature communications.

Reviewer #2:

Remarks to the Author:

All my questions have been answered, comments have been taken into account, and appropriate changes have been made to the text.

The article can be published in the current version.